# OrtDet: An Orientation Robust Detector via Transformer for Object Detection in Aerial Images

Ling Zhao [1] , Tianhua Liu [1], Shuchun Xie [2,*], Haoze Huang [1] and Ji Qi [1]

1    School of Geosciences and Info-Physics, Central South University, South Lushan Road, Changsha 410083, China
2    School of Traffic and Transportation Engineering, Changsha University of Science & Technology, Changsha 410114, China
*    Correspondence: xiesc@csust.edu.cn

**Abstract:** The detection of arbitrarily rotated objects in aerial images is challenging due to the highly complex backgrounds and the multiple angles of objects. Existing detectors are not robust relative to the varying angle of objects because the CNNs do not explicitly model the orientation's variation. In this paper, we propose an Orientation Robust Detector (OrtDet) to solve this problem, which aims to learn features that change accordingly with the object's rotation (i.e., rotation-equivariant features). Specifically, we introduce a vision transformer as the backbone to capture its remote contextual associations via the degree of feature similarities. By capturing the features of each part of the object and their relative spatial distribution, OrtDet can learn features that have a complete response to any direction of the object. In addition, we use the tokens concatenation layer (TCL) strategy, which generates a pyramidal feature hierarchy for addressing vastly different scales of objects. To avoid the confusion of angle regression, we predict the relative gliding offsets of the vertices in each corresponding side of the horizontal bounding boxes (HBBs) to represent the oriented bounding boxes (OBBs). To intuitively reflect the robustness of the detector, a new metric, the mean rotation precision (mRP), is proposed to quantitatively measure the model's learning ability for a rotation-equivariant feature. Experiments on the DOTA-v1.0, DOTA-v1.5, and HRSC2016 datasets show that our method improves the mAP by 0.5, 1.1, and 2.2 and reduces mRP detection fluctuations by 0.74, 0.56, and 0.52, respectively.

**Keywords:** object detection; rotation-equivariant; self-attention





## 1. Introduction

Object detectors for remote sensing images are designed to quickly and accurately search for target objects in images, such as vehicles, aircraft, playgrounds, and bridges. They are essential in traffic management, target detection (military), land use, and urban planning [1–3]. As remote sensing images are acquired with a top-view perspective, oriented object detectors use OBBs, which provide more accurate orientation information of the objects than the general object detectors [4–8] that use HBBs. This application enables better detections for objects with dense distributions, large aspect ratios, and arbitrary directions.

Many detectors designed for OBB tasks on remote sensing have reported promising results. In order to enhance the robustness and make the detector capable of detecting objects in arbitrary directions, most are devoted to learning the rotation-equivariant features of the objects [4,8–11]. As analyzed in the experiments of this paper, in the commonly used dataset of remote sensing, many categories not only have a small number of samples but also are concentrated in small angle intervals. Traditional detectors based on CNNs cannot learn rotation equivalence accurately because the convolutional kernels do not follow the directional changes of the objects [9,12]. Existing detectors can detect arbitrary orientation

objects by employing larger capacity networks that fit the feature expression of the object under different directions in the training set [13]. As shown in Figure 1a, we found that when the angle in the validation set is rotated to increase its distribution and training set differences, the performance of traditional detectors can differ by nearly 15 AP. Therefore, the robustness of the detector cannot be guaranteed. As shown in Figure 1b, the features maps extracted by traditional detectors are concentrated in the center of the object no matter how they are rotated, and it is difficult to learn the directional information of the object. It is crucial for the detector to encode the rotational equivalence features of the objects more accurately. For example, the CNNs have translation equivalence feature, so the presence of the object at any position does not have a significant impact.

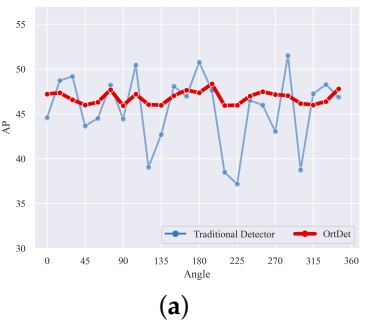

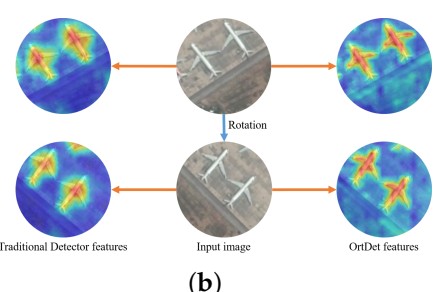

(**a**)                                                                    (**b**)

**Figure 1.** (**a**) The horizontal coordinate is the objects rotation angle in the validation set, the vertical coordinate is the AP, the red curve is our detector OrtDet, and the blue curve is the traditional detector. (**b**) Illustration of our method. The orange arrow represents the feature heat map, the blue arrow represents the rotation operation. Traditional detectors have difficulty in learning the orientation information of the objects, and their extraction features are focused on the center of the objects regardless of the rotation. Our OrtDet learned the orientation information of the object better.

Recently, to improve the ability of CNNs to encode the rotation equivalence of the object, group convolutions [14] have been proposed to extend the traditional CNNs to obtain the rotation equivalence features in larger groups, which produce feature maps with dedicated orientation channels to record features in different directions of the image [15–17]. The ReDet [13] builds on this approach by designing a rotation-invariant region of interest (RiRoI) structure to obtain features of the object in the directional channel and successfully confer the CNN models with rotation equivalence in detection tasks. However, such methods not only increase the computational complexity of the CNNs but also introduce a large amount of nonessential computations, e.g., most of the remote sensing image regions are composed of a background, and we do not need to obtain features of multiple directions in the background region. Cheng [18] et al. explicitly added a rotation-invariant regularizer to the CNN's features by optimizing a new loss function to force a tight mapping of the feature representations of the training samples before and after rotations to obtain rotation equivalence images. However, such methods force the CNN to learn high-dimensional image-level rotation equivariant/invariant features. In contrast, instance level rotation-equivalence features are more noteworthy in the object detection task.

In addition, the existing detectors [8,19–21] measure the precision using the mean average precision (mAP) metric. There is no specific metric for measuring the robustness of the object direction in the oriented object detection task. To a certain extent, the ability to learn more rotation equivalence features results in a higher mAP metric. However, the higher mAP does not equate to better rotation equivalence features of the object. The training and validation sets are usually divided randomly on the dataset, so there are independent and identically distributed [22–24] in direction distribution. Therefore, the detectors can achieve a high mAP by fitting the feature expression in each direction of the train set. It is equivalent to informing the detector in advance of the test set's object in the corresponding direction feature, and it cannot learn rotation equivalence features. In

particular, for remote sensing dataset bridges, courses and ports, and other objects, due to the limitation of the number of satellites and the limitation of the overhead perspective, the number of objects is not only small but also more than 50% of objects directions are concentrated in the range of $0 \pm 15°$. When the angle distributions of the training and test dataset are inconsistent, the mAP will be substantially reduced. Thus, the mAP metric can only reflect the accuracy of the detector for the current angle distribution dataset and does not reflect the robustness of the detector.

In this paper, we propose a novel OrtDet architecture for remote sensing images to improve robustness by learning the rotation equivalence features of the objects. Firstly, we introduce the vision transformer architecture as the backbone. As shown in Figure 1b, compared with the traditional detector, which focuses on the central part of the object regardless of how the object is rotated, and the extracted feature is very coarse, the OrtDet can regress the overall features of the object (e.g., nose, fuselage, wings, etc.) more finely and accurately, so we can identify the orientation of the object by its features, which is more suitable for the oriented object detection task. Unlike CNNs with fixed and limited perceptual fields, it relies on an intrinsic self-attentive mechanism as the main module to capture its remote contextual associations via the degree of feature similarity. The feature similarity region can change adaptively according to the object direction. As shown in Figure 2a, it can be seen that the feature similarity regions obtained by self-attention at each point are treated as nondifferentiated at the beginning of training. When the model is trained, as shown in Figure 2b, the model learns that the similarity region of the features at each point in the object converges adaptively from the full image range to the region where the object is located, and Figure 2c shows that this similarity region is consistent with the change in the object's orientation, thus demonstrating that this approach can better learn rotational equivalence features. This direction also follows the ViT [25] philosophy to reduce the "induction bias" while pursuing generalized features. Self-attention [26] computations have fewer inductive bias than CNNs, such as translation equivalence and localization, but it can still learn translation equivalence and scale equivalence features under certain datasets and supervised/self-supervised tasks [25,27], so it can also learn object rotation equivalence under the corresponding supervised tasks, i.e., oriented object detection tasks. Secondly, the transformer's architecture is a plain, nonhierarchical architecture that maintains a single-scale feature map [25]. We use the Tokens Concatenation Layer (TCL) layer, which reduces its spatial resolution with the increase in network depth and thus generates a multiscale features map.Then, we leverage the Feature Pyramid Network (FPN) [28] module to assign feature levels according to the scale of objects. To reduce the computational complexity of the model, we use the window mechanism and shift window mechanism to reduce the computational overhead of the model while maintaining the relationship between the windows of the images [29].

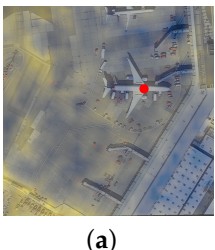 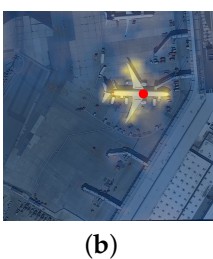 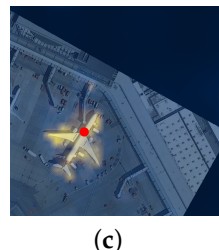

(**a**) (**b**) (**c**)

**Figure 2.** The relative importance of a single token (red point) with respect to the other tokens in the image is visualized. The shade of yellow color represents the degree of importance. (**a**) represents the beginning of the transformer's backbone training. (**b**) represents the state after the transformer's backbone is trained. (**c**) shows the object at different angles in the image.

In addition, we use the vertex offset method to accurately describe an oriented object by indirectly regressing the glide of each vertex offset of the object with the smallest HBBs

on each corresponding side. The principle is outlined in Section 3.4. This representation can further alleviate the instability of the regression's results due to the change in the object's direction. Furthermore, to quantitatively measure the ability of the detector to learn the object rotation equivalence features, we propose a new evaluation metric: mean Rotation Precision (mRP). It measures the fluctuation of the accuracy of the detector for the same dataset at different angle distributions. A smaller mRP means that the accuracy is less affected by the object direction in the dataset and is more robust. The effectiveness of the proposed method is demonstrated by extensive experiments on the remote sensing datasets DOTA-v1.0, DOTA-v1.5, and HRSC2016.

The significant contributions of this paper can be summarized as follows:

- We propose an Orientation Robust Detector for remote sensing images. It adaptively captures remote contextual associations via the degree of feature similarity. The rotation equivalence feature can be learned more accurately for objects at different angles.
- We use windowing mechanism and TCL strategy to reduce the computational complexity of transformer and generate multi-scale features. A more efficient and robust method is used to represent the oriented objects to reduce the regression's confusion problem.
- We propose a new metric, mRP, compared to the traditional mAP; it reflects the ability to learn object rotation equivalences by quantitatively describing the fluctuation of the accuracy in the test set for different object angle distributions.
- We carry out experiments on the DOTA-V1.0, DOTA-V1.5, and HRSC2016 datasets to demonstrate the effectiveness of our method, which involves significant improvements in detection performances as measured by both mAP and mRP metrics.

## 2. Related Work

### 2.1. Oriented Object Detection

Unlike most HBB object detectors [30–35], OBB object detectors [4,8,13,19,20] are more suitable for the overhead perspective of the remote sensing images. To detect objects in arbitrary directions, the method [36] determines OBBs by directly introducing angular parameters based on the Faster RCNN [31] regression center coordinates, width, and height. Rotated RPN anchors [37–39] with different angles, aspect ratios, and scales are defined manually to perform the regression of the object, while RRoI pooling is used to rotate the object features for normalization. However, the above method dramatically increases the extra calculation burden in the training and testing phases. The RoI-Transformer [4] interposes a light-weighting module between RPN and RCNN. A lightweight module is inserted to transform horizontal RoIs (HRoIs) into rotation RoIs (RRoIs), thus avoiding the generation of a large number of anchors and reducing the computational complexity. To avoid the problems of abrupt boundary variability and angular periodicity, the gliding vertex [19] first extracts the external horizontal bounding boxes of the object and then introduces scale factors on each side of the HBBs to locate the vertices of the OBBs. CSL [20] used angle classification instead of regression and designed soft labels to cope with the boundary problem, which achieved good results, but angle classification also led to an excessive amount of output parameters. R3Det [8] and S2A-Net [10] designed a feature refinement module using feature interpolation to achieve better spatial alignments with oriented objects. Some complex and uncommon methods have recently been developed to represent OBBs using the middle line [7] and polar coordinates [40]. Zhang et al. [41] propose a novel oriented object detector based on CNN with adaptive object orientation features. All the above methods aim to improve the representation of the OBBs or feature representation of the object. The former can address the loss of abruptness caused by the angle periodicity and the interaction between the length and width of the edges. The latter can extract the exact features of the corresponding rotated object.

### 2.2. Rotation Equivalence Networks

To alleviate the deficiency that CNNs have a poor representation of rotational equivalence features, the group convolution method [14] is proposed to incorporate a 4-fold approximation with respect to the rotational equivalence feature. HexaConv [17] used a hexagonal raster expansion relative to a 6-fold rotation equivalence and obtained rotation-equivalence features at more angles. The ReDet [13] approach builds on this by modifying the RRoI structure in the detector to allow the cyclic switching of the direction channels and the interpolation of the corresponding features to obtain the rotation-equivalence features of the object. STN [42] and DCN [43] are widely used in oriented object detection tasks by explicitly modelling the rotation variation in the detector. The RICNN [44] optimizes a new loss function by introducing regularization. Cheng [18] et al. explicitly add a rotation-invariant regulariser to the CNN's features by optimizing a new loss function to force a tight mapping of the feature representations to achieve rotational equivalence features. OSIm [45] combines the rotation-equivalence channel features constructed in the frequency domain with the original spatial channel features to cope with the problem. Although these methods approximate rotation-equivalence features, they all require many training samples and a large number of parameters. Secondly, these methods all focus on image-level rotation-equivalence features instead of instance-level rotation-equivalence features.

### 2.3. Transformer-Based Networks

In contrast to the above methods based on CNN backbone methods, our approach employs a transformer structure based on the self-attention mechanism [26] as the backbone of the detector. The advantage of the transformer lies in using self-attention for capturing global contextual information to establish a long-range dependence. ViT [25] applied a standard transformer directly to image classification by splitting the image into patches and not focusing on pixels, leading to very competitive results with CNNs. Since the architecture is a single-scale feature map with high computational complexity, it cannot be directly used as a backbone for detection tasks [27]. The swin transformer [29] was proposed to reduce the high number of computations by introducing the window and shift window mechanism. By limiting self-attention computation to the local window in the image, it is able to achieve higher accuracies than all existing CNN-based state-of-the-art methods. DETR [34] reduces the computational complexity of the transformer by combining the transformer and CNN, applying transformer architecture to object detection for the first time. Deformable DETR [46] borrows the idea of a variable convolution neural network that enables the transformer's ability to detect multiscale objects. Adaptive clustering transformers [47] further reduce the computational complexity of the model by clustering and accelerate the convergence process. SMCA-DETR [48] introduced a spatially modulated Gaussian mechanism to achieve fast convergence speeds compared with DETR. In the field of remote sensing, the transformer is also used in the field of object detection [49] and instance segmentatio [50,51]. In addition, the CRTransSar [52] structure has also been proposed for applications in SAR images for the object detection of ships. Zhang et al. [53] proposed a TRS structure to apply transformer for a remote sensing scene-classification task. The above methods use transformers as the backbone in the model, when combining them with CNNs to reduce their computational complexity, and there are no allocated supervised tasks for mining the rotational equivalence features of the detector.

## 3. Method

### 3.1. Overview

This section presents the principle of the proposed OrtDet to encode rotation-equivalence features and the box regression mechanism. The overall architecture of OrtDet is shown in Figure 3. Firstly, the image is split into tokens by patch partition and is fed into the backbone of the transformer's architecture. Then, it produces multiscale feature expressions relative to the image by the TCL layer. Secondly, the features corresponding to the smallest

HBBs of the object are extracted by RPN and RoI Align. Finally, the OBBs are determined by regressing the four vertices offset of the HBBs by the FC layer.

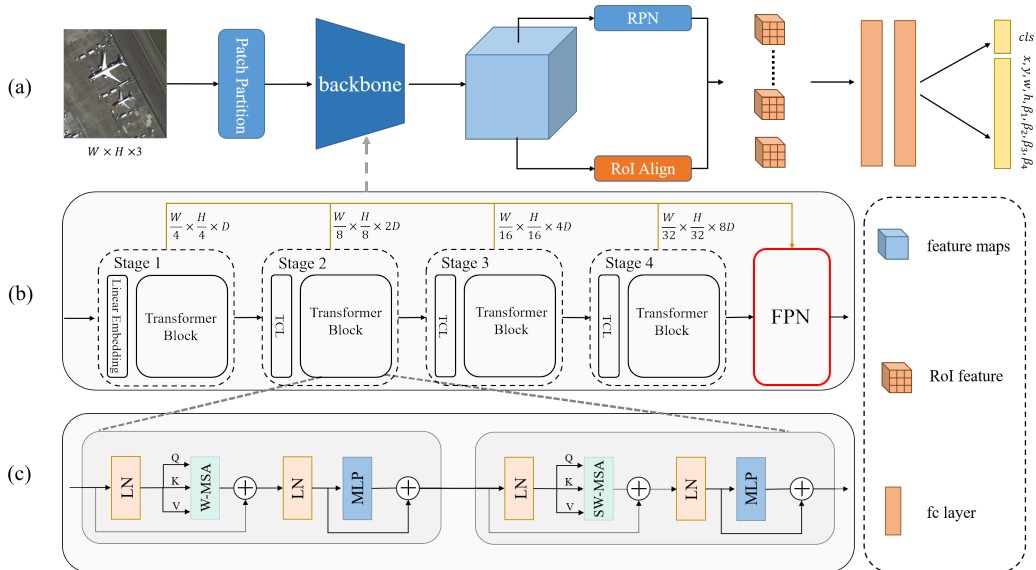

**Figure 3.** Overview of the proposed method. (**a**) Main structure of rotating object detection. (**b**) Main structure of Transformer model backbone. The brown arrows represent the features at each scale in the FPN. (**c**) Structure of the transformer block for feature extraction, where LN represents layer norm operations, W-MSA represents self-attention operations in the rule window, and SW-MAS represents self-attention operations in the offset window.

### 3.2. Feature Learning

We introduce the transformer architecture as the backbone. To enrich the spatial neighborhood information of each pixel and reduce the computation, input image $I$ is first divided into $H \times W$ patches, $x_i \in \mathbb{R}^{p^2 * C}, i \in \{1, \cdots, N\}$ by a patch partition module, where $P$ is the size of each patch, $C$ is the channel number of the image (in RGB image usually 3), and $N$ is the total number of patches (Figure 4). The Transformer uses a constant latent vector size $D$ via all of its layers, so the tokens are projected to an arbitrary channel dimension $D$ by using linear embedding layer $U$ and each patch is mapped to a token that enriches the feature's information. The token after patch partition is denoted by $z^0$, as shown in Equation (1).

$$z^0 = [x_1 U, x_2 U, \cdots, x_N U], U \in \mathbb{R}^{(p^2 \cdot C) \cdot D}, z^0 \in \mathbb{R}^{N \cdot D} \tag{1}$$

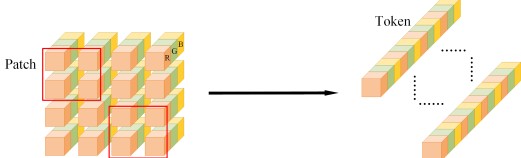

**Figure 4.** The original image is divided into nonoverlapping Patches(red boxes), and the original spectral features in the Patches are concatenated in the spatial dimension to be output as tokens of the original image. The dotted lines represent the rest of the Tokens that are omitted

Our model uses multi-head self-attention (MSA) [26] to find the relative importance of a single token with respect to the other tokens in the image, which is measured by calculating the attention weight between each token. To obtain the attention weight, the $Q$ (query), $K$ (key), and $V$ (value) vectors are generated by multiplying the element against three learned matrices $U_{QKV}$. (Equation (2)). The vector for Q, K and V dimensions is

denoted by $D_K$. Vector $Q$ and $K$ contain the similarity of each token in the image obtained by the dot product of $Q$ and $K$. Next, the output similarity matrix is weighted by multiplying the $V$ vector to obtain the final results. Since vectors $Q$, $K$, and $V$ are generated by tokens, the method is called self-attention.

$$[Q, K, V] = Z^i U_{QKV}, U_{QKV} \in \mathbb{R}^{D*3D_K} \tag{2}$$

To determine the relevance between tokens, the dot products of the $Q$ of the focused token and the $K$ of other tokens are obtained. The dot product's result is normalized by $D_K$ and fed into a softmax to obtain attention weight $AT$ (Equation (3)). The $AT$ determines the relative importance of every pair of tokens in the image.

$$AT = Softmax(\frac{QK^T}{\sqrt{D_K}}), AT \in \mathbb{R}^{N*N} \tag{3}$$

Each row of the $AT$ represents the relative importance of a single token with respect to the other tokens, so the $V$ of each token is multiplied by the output of the softmax. The result can be interpreted as the most closely associated tokens, thus extracting its corresponding feature. Since the highest score is usually used to represent itself, it is necessary to use $MSA$ to extract different relevant features.

$$ST = AT \cdot V \tag{4}$$

The MSA block computes the scaled dot-product attention separately for $N$ heads using the previous operation, but each of the different heads is projected to a different Q-K-V vector using a different $U_{QKV}$ matrix. The output of each of the $N$ heads is concatenated and projected to the specified feature dimension using a feedforward layer with learnable weights $W$.

$$MSA(z) = Concat(SA_1(z_i); SA_2(z_i), \cdots, SA_N(z_i)) \cdot W, W \in \mathbb{R}^{nD_k*D} \tag{5}$$

To further reduce the computational complexity of the transformer architecture, we adopt a windowing mechanism. Instead of performing self-attention with all the remaining tokens in the image, we divide the tokens into separate nonoverlapping windows by region, as shown in Figure 5a. Moreover, only the self-attention between each token in each window is calculated, as shown in Equation (6). $W\_MSA$ represent the $MSA$ operations in windows.

$$\hat{z}^l = W\_MSA(LN(z^{l-1})) + z^{l-1} \tag{6}$$

$$z^l = MLP(LN(\hat{z}^l)) + \hat{z}^l \tag{7}$$

Secondly , to prevent the restrictive isolation of tokens in each window, as shown in Figure 5b, the shift windows operation is introduced to allow information interactions between adjacent and nonoverlapping local windows. It significantly enhances the feature representation capability of the model, as shown in Equation (8). $SW\_MSA$ represent $MSA$ operations in shift windows. Each transformer block followed by a two-layer MLP and LayerNorm(LN) layer is applied before each MSA module and each MLP module, as shown in Equations (7) and (9).

$$\hat{z}^{l+1} = SW\_MSA(LN(z^l)) + z^l \tag{8}$$

$$z^{l+1} = MLP(LN(\hat{z}^{l+1})) + \hat{z}^{l+1} \tag{9}$$

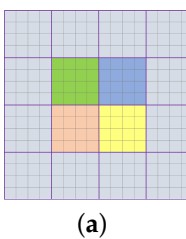

(**a**)

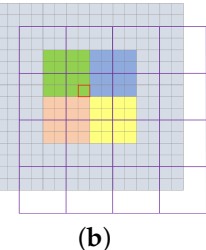

(**b**)

**Figure 5.** Self-attention in windows portioning (**a**) and Shift windows partitioning (**b**) operation. The orange, yellow, green and blue colors represent Tokens in each window in (**a**), the red boxed pixel in (**b**) can be used to perform a self-attention operation with the four adjacent pixels in the previous windows

### 3.3. TCL Layer

We use the TCL layer to transform nonhierarchical, single-scale features into hierarchical, multiscale features and conduct the task of multiscale object detection in remote sensing images. The detailed operation procedures are as follows.

Specifically, we perform a concatenation operation on the $2 \times 2$ neighbor tokens in the feature map, which reduce the feature scale of tokens to half of the original one and change the channel dimension from D-dimension to 4D-dimension. To prevent the dimension from growing too fast and to render the merged features smoother, we use linear layer $U_d$ to reduce the feature dimension from $4D$ to $2D$. (Equation (10))

$$z'_{\frac{j}{2},\frac{i}{2}} = Concat(z_{(i,j)}, z_{(i+1,j)}, z_{(i,j+1)}, z_{(i+1,j+1)}) \cdot U_d, U_d \in \mathbb{R}^{4D*2D}, z' \in \mathbb{R}^{\frac{N}{2}*2D} \qquad (10)$$

In stage 1, we do not use the TCL layer to reduce the feature scale because the original image input feature dimension is expanded to the D dimension using linear embedding. In stage 2, the feature scale is reduced from $\frac{W}{4} \times \frac{H}{4}$ to $\frac{W}{8} \times \frac{H}{8}$ using the TCL layer, and in stage 3 and stage 4, the feature scale is reduced to $\frac{W}{16} \times \frac{H}{16}$ and $\frac{W}{32} \times \frac{H}{32}$, respectively. As shown in Figure 6, the model is designed as a hierarchical pyramid structure via the TCL layer to produce a hierarchical feature representation. Stage 1 and Stage 2 downsampled the images by $4\times$ and $8\times$, respectively, which produced shallow features with strong spatial and geometric features, but lacked semantic features for the images. Stage 3 and Stage 4 downsampled the images by $16\times$ and $32\times$, respectively, which produced deep features that responded to high-level semantic features of the objects, but lacked spatial geometric features. Therefore, we use the FPN [28] module to fuse the shallow and deep features of the image and perform the object detection task independently in each level by top-down and skip connections. Thus, we can satisfy multiscale object detection features in remote sensing images.

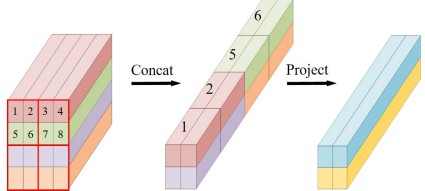

**Figure 6.** To describe this operation more visually, we use different colors for the features of different rows. The original tokens in the red box are concatenated with their feature vectors in the spatial dimension(e.g. 1, 2, 5 and 6 tokens), then projected to the specified feature dimension.

### 3.4. Rotated Object Box Regression

The direct regression of the angle parameters is challenging for the detector. Therefore, we use the regression of HBB corner point offsets to determine the OBB. An intuitive illustration of the method is depicted in Figure 7. We first calculate the HBB (black box $B_h$

in Figure 7) of the OBB (blue box $B_o$ in Figure 7). $B_h$ is represented using the conventional methods $(x, y, h, w)$, with $(x, y)$ representing the center point and $(h, w)$ representing the width and height, respectively. In addition to the HBB's representation, four additional parameters $(\beta_1, \beta_2, \beta_3$ and $\beta_4)$ are introduced to describe the OBB of the object. The formula for $\beta_i, i \in \{1, 2, 3, 4\}$ is shown in (Equation (11))

$$
\begin{aligned}
\beta_i &= \frac{l_i}{w}, i \in \{1, 3\} \\
\beta_i &= \frac{l_i}{h}, i \in \{2, 4\}
\end{aligned}
\tag{11}
$$

where $l_i$ represents the offset of each corner point in the OBB from the corresponding corner point in the HBB. The normalization operation allows the model to converge quickly to the target object.

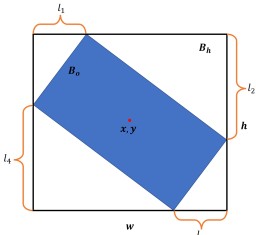

**Figure 7.** First, determine black box $B_h$ (x, y, w, and h) of the object's border, and the red dot represents the coordinates of the object center point. Then calculate the offset of each corner point($\beta_1, \beta_2, \beta_3$ and $\beta_4$) in the HBB to describe the blue box, $B_o$, of the object.

The input image is first fed into the backbone to extract features and generate HBBs proposals with RPN. Then, the regional features extracted via RoIAlign on the proposals are passed through the FC layers to generate final results, including a HBB (x, y, h, and w) and four variables ($\beta_1, \beta_2, \beta_3$, and $\beta_4$) defining the OBB. The loss function of the model is similar to that of the Faster RCNN [31]. The difference in the regression loss of object bounding box $L_{reg}$ is calculated according to Equation (12).

$$
\begin{aligned}
L_{reg} &= \lambda_1 L_h + \lambda_2 L_\beta \\
L_\beta &= \sum_{i=0}^{4} smooth_{L_1}(\beta_i - \widehat{\beta_i})
\end{aligned}
\tag{12}
$$

where $L_h$ is the HBBs loss of the object, which is calculated in the same way as in the Faster RCNN, and $\widetilde{\beta_i}$ represents the true offset of each corner point of the rotating border. $\lambda_1$ and $\lambda_2$ are hyperparameters to balance the importance of each part of the loss.

### 3.5. Rotated Precision Metric

To address the defect that the mAP does not accurately measure the model's ability to learn the rotation equivalence of the object, we propose a new mean rotational precision (mRP) as an evaluation metric. Firstly, all images in test set $I$ containing $K$ categories are randomly rotated by $N$ sets of different angles to form an extended test set $\{I^1, I^2, I^3, \ldots, I^n\}$. Then, the accuracy of $AP_j^i$ is evaluated for each object class in the test set $I_j^i$ ($1 \le i \le n, 1 \le j \le k$), and the variance value, $RP_j$, of $AP_j^i$ accuracies for each category in the rotated test set is calculated as follows:

$$
RP_j = \frac{\sum_{i=1}^{n}(AP_i^j - M_j)^2}{k}
\tag{13}
$$

where $M_j$ represents the average accuracy of each category under different angle distributions.

$$M_j = \frac{\sum_n^{i=1} AP_i^j}{n} \tag{14}$$

Finally, the mRP results are obtained by averaging various types of objects:

$$mRP = \frac{\sum_{j=1}^{k}(RP_j - M)^2}{k} \tag{15}$$

where $M$ represents the average of the various types of $RP_j$ accuracies.

$$M = \frac{\sum_{j=1}^{k} RP_j}{k} \tag{16}$$

$RP_j$ represents the degree to which the accuracy is affected by the angle for each category, with smaller values representing the model's ability to learn the rotation equivalence feature of the object better. mRP is the robustness of the mAP for each dataset, and a smaller mRP means a more robust model, leading to fewer errors due to changes in the object's direction.

## 4. Experiments and Analysis

This section evaluates the proposed OrtDet on three public datasets designed for remote sensing oriented object detection task. We first introduce the dataset used in the experiment (Section 4.1). Then, we describe the implementation details and hyperparameter design (Section 4.2). Finally, we analyze our model and present our experimental results on different datasets (Sections 4.3–4.5).

### 4.1. Dataset Description

The experiments evaluate the proposed OrtDet on three publicly available datasets designed for remote sensing oriented object detection.

**DOTA-v1.0** [36] is one of the largest datasets for oriented object detection in the remote sensing field released by Wuhan University in China. It contains 2806 remote sensing images with sizes ranging from 800 × 800 to 4000 × 4000 and containing 188,282 object instances and 15 object categories, which includes the following: Plane (PL), Baseball diamond (BD), Bridge (BR), Ground track field (GTF), Small vehicle (SV), Large vehicle (LV), Ship (SH), Tennis court (TC), Basketball court (BC), Storage tank (ST), Soccer-ball field (SBF), Roundabout (RA), Harbor (HA), Swimming pool (SP), and Helicopter (HC).

**DOTA-v1.5** is a new dataset released in the 2019 DOTA Challenge that adds to the DOTA-v1.0 dataset, mainly by adding a new category: Container Crane (CC) and giving many small object instances smaller than 10 pixels for specific annotations, especially on the SV category for more detailed annotation. Compared to DOTA-v1.0, the DOTA-v1.5 dataset is more challenging, but the model training can be more stable.

**HRSC2016** [54] is a challenging ship detection dataset with OBBs annotation produced by Northwestern Polytechnical University in China in 2016, which contains 1061 aerial images that range in size from 300 × 300 to 1500 × 900. It contains 436 training sets, 181 validation sets, and 444 test sets. Similarly to DOTA, we used separate training and validation set. All images were resized to (800,512) without changing the aspect ratio, and a random horizontal flip was applied during training.

For a fair comparison, we process the dataset similarly to the baseline method. When evaluating the mAP metric in the DOTA dataset, the training and validation sets are used for training, the test set is used for testing, and the accuracy is evaluated using the official code. To calculate the rotation precision of the mRP, we need to rotate the image and label information simultaneously to evaluate the accuracy of the dataset at different angular distributions. As we do not have access to the label information of the test set, we only use the training set for training and rotate the validation set for testing. We cropped the original

image to 1024 × 1024 for image processing, set the step size to 824, and used 0 to paddle images smaller than 1024 after cropping. The sample distribution in each category in the DOTA-v1.5 dataset is shown in Figure 8a. The number of samples in each category is shown in Figure 8b. As depicted in the figure, it can be seen that in the remote sensing dataset, ground nonfixed objects mainly include targets such as vehicles, planes and ships, and ground-fixed objects mainly include targets such as courts, bridges, and ports. The number of objects for ground nonfixed objects is usually much larger than that of ground-fixed objects. Furthermore, the object directions of ground-fixed objects are concentrated in a few angle intervals. On the other hand, due to the arbitrary nature of the direction in ground nonfixed objects, the object's directions are evenly distributed in each interval, so it is consistent with the previous analysis of this paper, which proves that the proposed model has a wide range of application scenarios in the field of remote sensing.

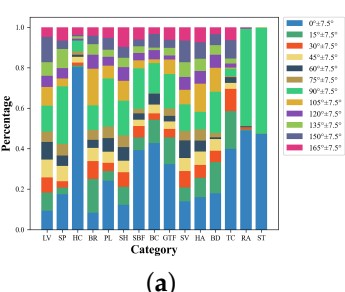 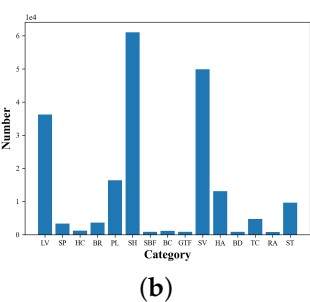

(**a**)  (**b**)

**Figure 8.** (**a**) The direct distribution of each category in the DOTA-V1.0 dataset by cropping the images at each angle. (**b**) The number of samples in each category after cropping the images in the DOTA-V1.0 dataset.

### 4.2. Implementation Details and Hyperparameter Design

The proposed method is mainly experimented on the RTX3090 using the MMDetection framework [55]. The initial learning rate is $7.5 \times 10^{-3}$ with a batch size of 4 for training and divided by 10 in each learning rate decay step. For the DOTA-v1.0 and DOTA-v1.5 datasets, the training epoch is set to 12, and the learning rate decayed at 8 and 11 epochs. We cropped the original image to 1024 × 1024 with a step size of 824. The data augmentation strategy is used to balance the categories in the training sample. For the HRSC2016 dataset, the training epoch was set to 36, and the learning rate decayed at 24 and 33 epochs. In all experiments, the network is trained by the SGD optimizer with momentum and weight decay set to 0.9 and $5 \times 10^{-4}$, respectively. The window size is set to $7 \times 7$ by default. The parameter, D, of the channel number is set to 96, and the expansion layer of each MLP is 4 for all experiments (e.g., the MLP input dimension is 96 and, the hidden layer's dimension is 384 ). In multi-head self-attention, the number of heads gradually increased as the transformer block deepened. In stage 1, stage 2, stage 3 and stage 4, the number of heads is 3, 6, 12, and 24, and the numbers of layers is set to 2, 2, 6, and 2, respectively. In addition, the hyperparameters $\lambda_1$ and $\lambda_2$ in Equation (12) are both set to 1. Data augmentation was performed by only using random horizontal and vertical flips. When evaluating the mRP's metrics, to ensure the authenticity of the results, we rotate the validation set by multiple angles to ensure that the operation does not change the size of the object, and we use a rotation followed by a cropping sequence for both the images and the annotation files. Moreover, we do not use rotated augmentation in the training process for the mRP metric.

### 4.3. Detection Precision

We compare the OrtDet model proposed in this paper with representative state-of-the-art methods on the three benchmark dataset in Tables 1–3. We indicate in boldface and highlight the best results on each column for the different methods. Our method shows the best performance on either dataset, achieving 74.09% mAP on the DOTA-v1.0 dataset, 64.4% mAP on DOTA-v1.5, and 89.6% mAP on the HRSC2016 dataset. In addition,

compared to the same vertex offset regression of the Gliding Vertex method, our OrtDet, can significantly improve the accuracy for small objects categories, such as Large Vehicle, Small Vehicle, Ship, and to a certain extent for other categories. The reason is that small objects are more sensitive to angular regression, and the slight deviation for small objects is more likely to reduce the IoU between the ground truth and the regression bounding box. In contrast, our OrtDet can obtain rotation-equivalence features better than CNNs, so the improvement in accuracy for such objects is noticeable. Moreover, in the DOTA-v1.5 dataset, the Small Vehicle category incorporates a large number of labels below 10 pixels. Thus, vertex offset regression is unsuitable for describing such tiny rotating objects, resulting in lower accuracies for the Small Vehicle category than other oriented object detection models. However, our detector can significantly alleviate this deficiency.

**Table 1.** Mean average precision (%) of the different methods for the DOTA-v1.0 dataset.

| Methods | PL | BD | BR | GTF | SV | LV | SH | TC | BC | ST | SBF | RA | HA | SP | HC | mAP |
|---|---|---|---|---|---|---|---|---|---|---|---|---|---|---|---|---|
| RRPN [37] | 88.52 | 71.20 | 31.66 | 59.30 | 51.85 | 56.19 | 57.25 | 90.80 | 72.84 | 67.38 | 56.69 | 52.84 | 53.08 | 51.94 | 53.58 | 61.01 |
| RoI-Transformer [4] | 88.64 | 78.52 | 43.44 | **75.92** | 68.81 | 73.68 | 83.59 | 90.74 | 77.27 | 81.46 | 58.39 | 53.54 | 62.83 | 58.93 | 47.67 | 69.56 |
| Gliding-Vertex [19] | 88.58 | **84.94** | 49.87 | 73.90 | 70.80 | 75.62 | 79.21 | 90.60 | 78.06 | **86.26** | 58.11 | **70.11** | 75.88 | 72.81 | 47.82 | 73.51 |
| $R^3Det$ [8] | **89.59** | 81.17 | 50.53 | 66.10 | 70.92 | **78.66** | 78.21 | 90.81 | **85.26** | 84.23 | **61.81** | 63.77 | 68.16 | 69.83 | **67.17** | 73.74 |
| OrtDet (Ours) | 89.02 | 84.82 | **50.68** | 71.45 | **72.06** | 74.98 | **87.64** | **90.85** | 78.76 | 85.81 | 58.97 | 68.95 | 67.29 | 70.49 | 59.61 | **74.09** |

**Table 2.** Mean average precision (%) of the different methods for the DOTA-v1.5 dataset.

| Methods | PL | BD | BR | GTF | SV | LV | SH | TC | BC | ST | SBF | RA | HA | SP | HC | CC | mAP |
|---|---|---|---|---|---|---|---|---|---|---|---|---|---|---|---|---|---|
| RetinaNet-O [33] | 71.4 | 77.6 | 42.1 | 64.7 | 44.5 | 56.8 | 73.3 | 90.8 | 76.0 | 60.0 | 46.9 | 69.2 | 59.6 | 64.5 | 48.1 | 0.8 | 59.2 |
| FR-O [36] | 71.9 | 74.5 | 44.5 | 59.9 | 51.3 | 69.0 | 79.4 | **90.8** | **77.4** | 67.5 | 47.7 | 69.7 | 61.2 | 65.3 | **60.5** | 1.54 | 62.0 |
| Mask R-CNN [56] | 76.8 | 73.5 | 49.9 | 57.8 | 51.3 | 71.3 | 79.7 | 90.5 | 74.2 | 66.0 | 46.2 | 70.6 | 63.1 | 64.5 | 67.8 | 9.42 | 62.7 |
| Gliding Vertex [19] | 72.1 | 82.4 | 47.5 | **73.4** | 34.6 | 68.0 | 80.3 | 90.7 | 71.1 | 69.1 | **49.8** | 68.1 | **68.5** | 66.9 | 57.8 | **14.6** | 63.3 |
| HTC [57] | **77.8** | 73.7 | **51.4** | 64.0 | **51.5** | **73.3** | 80.3 | 90.5 | 75.1 | 67.3 | 48.5 | 70.6 | 64.8 | 64.4 | 55.9 | 5.15 | 63.4 |
| OrtDet (Ours) | 72.3 | **84.5** | 48.7 | 71.1 | 42.1 | 73.2 | **80.7** | 90.7 | 71.2 | **69.5** | 47.8 | **71.3** | 67.9 | **67.4** | 58.5 | 13.8 | **64.4** |

**Table 3.** Mean average precision (%) and mean rotation precision of the different methods for the HRSC2016 dataset.

| Methods | RRPN [37] | RoI-Transformer [4] | Gliding-Vertex [19] | OrtDet (Ours) |
|---|---|---|---|---|
| mAP | 79.6 | 86.2 | 87.4 | **89.6** |
| mRP | 2.55 | 2.37 | 2.28 | **1.76** |

*4.4. Rotation Equivalence*

In order to quantitatively assess the ability of baseline and the OrtDet to learn object rotation-equivalence features, the experiments are conducted to measure the extent to which the accuracy was affected by the angles distribution of each category in three datasets using the RP metric. The mRP is further used to quantitatively measure the overall fluctuation degree of the detector. The results are shown in Tables 3–5. We indicate in boldface and highlight the best results on each column for the different methods.

**Table 4.** Mean rotation precision (%) of the different methods for the DOTA-v1.0 dataset.

| Methods | PL | BD | BR | GTF | SV | LV | SH | TC | BC | ST | SBF | RA | HA | SP | HC | mRP |
|---|---|---|---|---|---|---|---|---|---|---|---|---|---|---|---|---|
| FR-O [36] | 0.28 | 3.75 | 2.06 | 3.98 | 3.37 | 2.18 | 0.47 | 5.07 | 4.20 | 2.58 | 4.18 | 2.05 | 4.72 | 2.95 | 8.79 | 3.45 |
| HTC [57] | 0.26 | 1.96 | 1.71 | 3.80 | 3.60 | 2.10 | 0.30 | 2.09 | 4.26 | 2.05 | 4.04 | 2.32 | 3.28 | 2.44 | 8.84 | 2.87 |
| RoI-Transformer [4] | 0.42 | 2.51 | **1.65** | 2.82 | 2.88 | 2.62 | **0.23** | 0.22 | 4.15 | 2.27 | 2.92 | 2.40 | 3.91 | 2.30 | 10.2 | 2.82 |
| $R^3Det$ [8] | 0.34 | **1.50** | 2.00 | 4.08 | 2.28 | 1.55 | 0.26 | 3.03 | 5.24 | 1.71 | 2.63 | **1.63** | 3.05 | 2.13 | 10.22 | 2.77 |
| Gliding-Vertex [19] | 0.13 | 3.63 | 1.83 | 2.97 | 2.70 | 0.66 | 0.35 | 0.21 | 3.20 | 0.84 | 2.92 | 2.23 | 3.82 | 3.75 | 7.64 | 2.38 |
| OrtDet (Ours) | **0.12** | 2.84 | 2.18 | **2.23** | **2.25** | **0.65** | 0.31 | **0.09** | **1.23** | **0.25** | **1.02** | 1.98 | **2.83** | **1.76** | **5.06** | **1.64** |

**Table 5.** Mean rotation precision (%) of the different methods for the DOTA-v1.5 dataset.

| Methods | PL | BD | BR | GTF | SV | LV | SH | TC | BC | ST | SBF | RA | HA | SP | HC | CC | mRP |
|---|---|---|---|---|---|---|---|---|---|---|---|---|---|---|---|---|---|
| FR-O [36] | 0.66 | 3.07 | 1.53 | 3.73 | 3.15 | 2.23 | 0.27 | 5.47 | 5.40 | 0.41 | 3.17 | 1.54 | 4.86 | 0.77 | 8.23 | **0.74** | 2.83 |
| Mask R-CNN [56] | 0.57 | 2.46 | 1.54 | 3.74 | 2.53 | 1.78 | 0.16 | 4.76 | 4.78 | 0.33 | 3.58 | 2.15 | 2.47 | 0.82 | 9.49 | 1.68 | 2.68 |
| HTC [57] | 0.36 | 2.77 | **0.87** | 3.51 | 1.93 | 0.97 | 0.49 | 4.21 | 4.03 | 0.36 | 3.94 | 1.61 | 2.94 | 1.48 | 7.92 | 3.06 | 2.53 |
| RoI-Transformer [4] | 0.40 | 3.00 | 1.44 | 2.82 | 1.60 | 0.90 | 0.53 | 1.77 | 3.78 | 0.48 | 2.44 | 3.69 | 1.14 | 8.65 | 3.85 | 2.41 |
| Gliding-Vertex [19] | 0.67 | 2.68 | 2.27 | 3.13 | **0.30** | 0.94 | 0.07 | 1.74 | 3.14 | 0.21 | 1.29 | **1.03** | 2.25 | 0.95 | 7.73 | 1.82 | 1.89 |
| OrtDet (Ours) | **0.23** | **1.35** | 1.23 | **2.11** | 0.76 | **0.80** | **0.05** | **0.63** | **1.60** | **0.12** | **0.72** | 1.11 | **1.58** | **0.90** | **5.01** | 0.99 | **1.20** |

The experimental results in Tables 5 and 6 show that the RP of certain categories for all the detectors are significantly smaller than others, such as aircraft, ships, and vehicles, for which the directional characteristics are apparent. The accuracy variation of plane and ship with their angular distribution is further shown in Figure 9. The distribution of these categories in the training set is shown in Figure 8a. As these categories are nonground fixed objects, they can be distributed at any angle in the overhead perspective of the remote sensing image. Hence, their distribution in each angle interval in the dataset is approximately balanced. Figure 10a,b show the plane and ship categories distribution on each angle interval in detail. In addition, the number of objects in these categories is much larger than the number of objects in other categories from Figure 8b. Thus, the CNN-based backbone has sufficient data for learning the features of these objects in different directions. Thus, it does not need to learn the rotation-equivalence features of the object and can still accurately regress the OBBs of these categories. The OrtDet model can further reduce RP fluctuations in these categories by learning rotation-equivalence features.

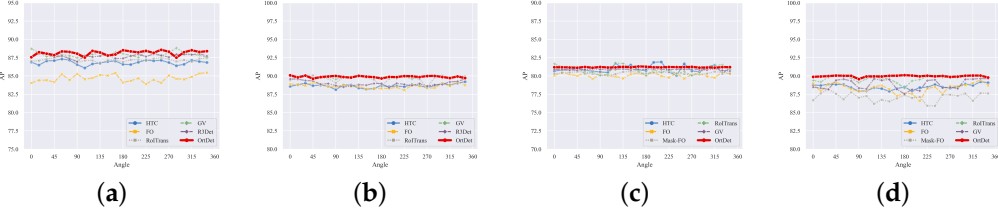

(**a**)        (**b**)        (**c**)        (**d**)

**Figure 9.** (**a**,**b**) show the fluctuation of the detection accuracy of the ship and plane categories in the DOTA-V1.0 dataset with the rotation angle of the dataset, respectively. (**c**,**d**) show the fluctuation of the detection accuracy of the ship and plane categories in the DOTA-V1.5 dataset with the rotation angle of the dataset, respectively.

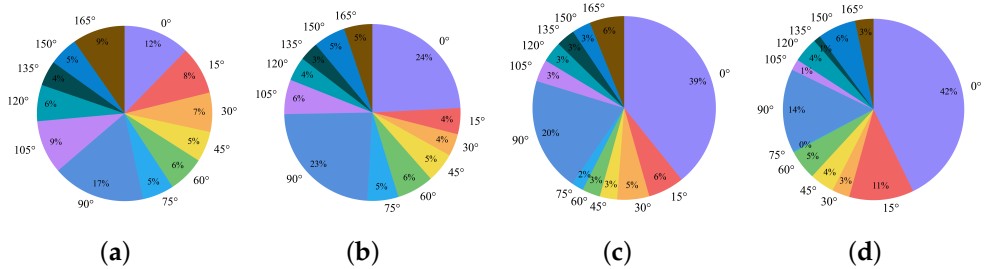

(**a**)        (**b**)        (**c**)        (**d**)

**Figure 10.** (**a**–**d**) represent the distribution of ship, plane, soccer-ball fields and basketball courts in each angle interval, respectively. We count the number of objects belonging to each angle interval from 0° to 165° in 15° intervals. From the figure, we can find that the distribution of plane and ship in each angle interval is much more balanced than that of basketball courts and soccer-ball fields.

**Table 6.** Performance comparisons of the backbone on OrtDet.

| Methods | Backbone | DOTA-v1.0 | | DOTA-v1.5 | | Size (Mb) |
|---------|----------|-----------|-----|-----------|-----|-----------|
| | | mAP | mRP | mAP | mRP | |
| OrtDet | ResNet | 70.89 | 2.38 | 59.89 | 1.75 | 60.4 |
| | Transformer | 72.32 | 1.64 | 61.50 | 1.19 | 52.9 |

For the roundabout and storage tank categories, although they are distributed within a specific range of angles, their accuracy does not fluctuate particularly significantly regardless of the rotation of the dataset due to the approximate circular shape of the objects. The main reason for the slight change is that the detector's result is always horizontal no matter how the circular object is rotated. However, to make the labels follow the rotation of the object, we rotate the ground truth of the dataset accordingly, thus resulting in a slight deviation of the result of the circular object from the ground truth after rotation.

The categories Basketball-court, Soccer ball field, Ground track field, and Harbor are ground-fixed objects. The angles do not change flexibly in the overhead perspective of remote sensing images, resulting in a relatively concentrated distribution of the object's angles. For example, the angle of the basketball court category in Figure 10c is concentrated around 0° and 90°, and the training samples of these categories are tiny compared with those of the plane, ship, and vehicle, as shown in Figure 8b. Therefore, the detector in the baseline does not has sufficient data to learn the features of these categories in other directions. Therefore, when the object angles of these categories in the validation set are inconsistent with the training data, the accuracy will drop sharply, proving that they cannot really learn the rotation equivalence feature. Contrarily, the accuracy of the OrtDet is much less affected by the angle distribution than that of the baseline. The model can rely on training data with a more concentrated angle distribution to obtain robustness results in the validation set with a completely different angle distribution. Figure 11 shows that the experimental results demonstrate that the method is much more capable of learning rotation-equivalence features than the baseline. As depicted in Figure 12, we visualize and evaluate the Basketball-Court, Helicopter, Plane, Soccer-ball-field, and Tennis-Court categories using a confidence threshold of 0.7. The OrtDet model proposed in this paper does not change significantly with a change in the object's direction when the accuracy is high, while baseline methods are prone to missing detections and false detections due to the change in the object's angle.

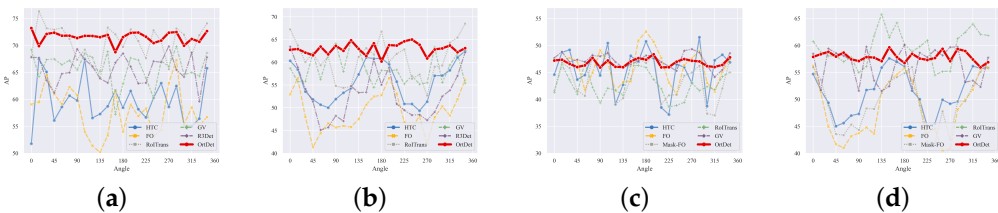

|  (a)  |  (b)  |  (c)  |  (d)  |

**Figure 11.** (**a**,**b**) show the fluctuation of the detection accuracy of the Soccert-ball-field and Basketball-court categories in the DOTA-V1.0 dataset with the rotation angle of the dataset, respectively. (**c**,**d**) show the fluctuation of the detection accuracy of the Soccert-ball-field category and the Basketball-court category with the rotation angle of the dataset in the DOTA-V1.5 dataset, respectively.

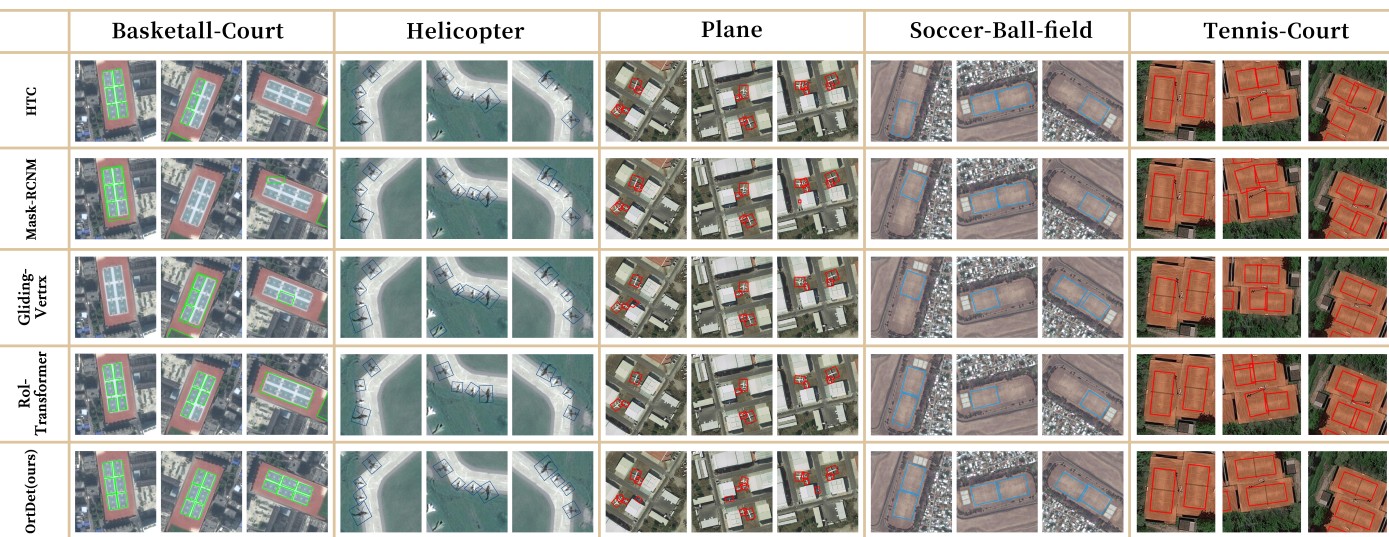

**Figure 12.** Qualitative comparison with baseline methods in detecting the identical objects categories of different orientations in the DOTA-v1.5 dataset by rotating an input image with different angles. The bounding box is the result of the detection for different categories. The OrtDet architecture is subject to much smaller angular images of the object than the baseline.

Figure 13 shows that in the DOTA-v1.0 and DOTA-v1.5 datasets, the OrtDet has a much smaller overall fluctuation than baselines. At the same time, the mAP of each angular distribution is slightly higher or similar to that of the baseline. Therefore, the OrtDet has a more significant advantage in encoding rotation equivalence features, and its learning of rotation equivalence features makes the detector more robust. In particular, in remote sensing images, due to the limitation of the satellite trajectory and overhead perspective, generating a large number of training samples with a more uniform angle distribution for fixed ground features is challenging. Thus, the robust detector is more beneficial to the task of the oriented detector in remote sensing images.

*4.5. Ablation Study*

In this section, we perform a series of ablation studies on the DOTA dataset to validate the effectiveness of the proposed method in this paper.

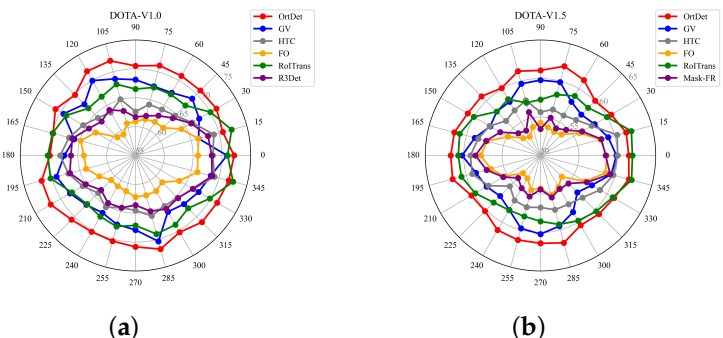

**Figure 13.** A higher roundness of the curve indicates a higher robustness of the detector, and a smaller distance between the curve and the outer circle indicates higher accuracies. (**a**) Variation of the detection accuracy with the rotation angle of the validation set on the DOTA-v1.0 dataset. (**b**) The variation of the detection accuracy with the rotation angle of the validation set on the DOTA-v1.5 dataset.

**Transformer Architecture**. To verify the effectiveness of the transformer architecture, we compared the ResNet backbone based on CNN. The experimental results are shown in Table 6. Compared with the ResNet, the transformer architecture can obtain a higher mAP

with further reductions in the number of parameters and significantly reduce the degree of mRP fluctuations. Thus, the transformer can better learn rotation equivalence features, and the rotation equivalence features of the object can improve the accuracy.

**Vertex offset**. In order to verify that the vertex offset method can accurately regress the OBBs, we compare the vertex offset method used in OrtDet with the most commonly used direct regression of the object angle method. The experimental results are shown in Table 7. The direct regression of the object angle will reduce the accuracy due to the imprecision of the angle regression. In contrast, the method of angle vertex offsets will slightly increase the number of parameters of the model due to the regression. However, this method can reduce the deviation of the regression results due to the change in the object angle based on the acquisition of the object rotation equivalence features and further improve the accuracy of the OBBs. Moreover, this method can improve the accuracy and reduce the fluctuation of the results.

**Table 7.** Performance comparisons of the box regression on OrtDet.

| Methods | Box Reg | DOTA-v1.0 | | DOTA-v1.5 | | Size (Mb) |
|---|---|---|---|---|---|---|
| | | mAP | mRP | mAP | mRP | |
| OrtDet | RBox reg | 71.88 | 2.04 | 61.33 | 1.88 | 50.8 |
| | Vertex offset | 72.32 | 1.64 | 61.50 | 1.19 | 52.9 |

## 5. Conclusions

In this paper, we propose the Orientation Robust Detector to improve the robustness relative to the object's direction in the oriented object detection task. Introducing the transformer's architecture alleviates the deficiency observed in traditional CNNs in learning object rotation equivalence. Moreover, its accuracy is robust relative to inconsistent angle distribution problems in the test set with the training set and/or the problem where data are insufficient in each angle range. Hence, the false and missed detections caused by the object direction variations are significantly reduced. We used the TCL layer to generate the transformer's pyramidal feature hierarchy to address the large-scale differences of objects in remote sensing. We also reduced the computational complexity of the model by introducing a windowing mechanism and a shift windowing mechanism. In addition, we predict the HBBs of the object and the relative gliding offset to represent the OBBs, which avoids the deviation of the OBBs regression due to the change in the object's direction. Finally, the mRP metric is proposed to quantitatively reflect the degree of accuracy fluctuation when the object orientation of the dataset changes as a measure of the model's learning ability for object rotation equivalence features. Experimental results fully demonstrate the effectiveness and robustness of our OrtDet. Nevertheless, the windowing mechanism introduced to reduce the complexity of the model leads to a limited interaction range between image regions. Our future work will focus on expanding the image's interaction range in the transformer architecture as much as possible while being computationally tractable.

**Author Contributions:** Conceptualization, L.Z.; Methodology, T.L.; Formal analysis, S.X.; Writing—original draft, L.Z.; Writing—review and editing, T.L., H.H. and J.Q.; Visualization, H.H.; Project administration, S.X., L.Z.; Funding acquisition, S.X. All authors have read and agreed to the published version of the manuscript.

**Funding:** This research was funded by National Natural Science Foundation of China(42171458); Natural Science Foundation of Hunan Province, China(2021JJ30818); Supported by the Young Teacher Development Program of Changsha University of Science and Technology(2019QJCZ006)

**Data Availability Statement:** The data that support the findings of this study are available from the author upon reasonable request. This data is available at https://captain-whu.github.io/DOTA/index.html (10 October 2022).

**Conflicts of Interest:** The authors declare no conflict of interest.

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
