# Peer review of "OrtDet: An Orientation Robust Detector via Transformer for Object Detection in Aerial Images"

_remotesensing, doi:10.3390/rs14246329_

Round 1

Reviewer 1 Report

This paper tackles the well known problem of making CNNs rotational invariant. There are several approaches to this such as data augmentation (ie training on many rotations of the object) and mapping the problem into a different coordinate system ie polar.

This paper uses the approach of feature extraction prior to the CNN. The general approach is well known but the implementation appears to be sound and produces competitive results compared with techniques. 

The English has many mistakes but these are mainly minor. They read as if the authors just need to take a little more care in checking the text, rather than any big gap in their knowledge. It should still be checked my native English speaker. 

Author Response

Thank you for your positive and constructive comments and suggestions on our manuscript. We have uploaded the point-by-point response as a PDF attachment.

Reviewer 2 Report

Summary:

In this work, the authors proposed a robust orientation detector to tackle the challenges of detecting oriented objects. The proposed method adapts transformers as backbone networks to learn rotation-equivariant features. The author also adapts TCL strategy for detecting objects at different scales and use relative gliding offsets to represent oriented bounding boxes to avoid angle regression. The proposed method is reasonable and straightforward, and the authors provided comprehensive experiments to support their claim. 

Strength:

1. the paper is well-written and easy to read and follow. 

2. The main novelty of this paper is using transformer to learn contextual associations and relative spatial distribution, which would improve the performance of detecting orientated objects.

3. The author claim transformer would be better than CNNs to learn the oriented features. Their claim is logical, and the author has provided thorough experiments to support their theory. 

4. In the experiment session, the author provided a new metric to evaluate the model's ability to learn the rotation-equivariant feature, which is a good complement approach for evaluating model performance on this task.

5. The author provided comprehensive analysis and experiments on multiple datasets to show the effectiveness of the proposed method.

Weakness:

1. The main weakness would be novelty and originality. The technical innovation of this paper is weak as most of the core modules and methodology (e.g transformers, TCL, orientated box representation) are presented in previous works. 

2. Although the transformer backbone has shown better performance than the resnet backbone. However, the benefits of using transformers are not large according to the ablation study. Moreover, it would be good to provide more comparisons to other backbone networks to further illustrate the advantage of using the transformer. 

Overall judgment:

Although lacking technique innovation in this work, I think it is a good attempt to use the transformer to learn the orientation-agnostic features and improve performance in detecting oriented objects. The proposed method is reasonable for the task and straightforward. Moreover, the authors' experiments and analyses have shown the effectiveness of their proposed method. 

Author Response

(The authors gave the same response as above.)

Reviewer 3 Report

1.     The FPN module in line 258 is not explained. Does the shallow and deep feature intermingling mentioned here refer to the fusion between the feature maps output from each stage? It is not reflected from the structure figure 3. A more detailed explanation is needed in the paper.

2.     The necessary parameters in the network structure should be supplemented, such as the parameters of multi head attention and the dimension of MLP input and output.

3.     How does the Fig. 1 b) show your OrtDet better learn the orientation information of the object?

4.     It would be beneficial to show examples of the datasets used by the authors for the article.

5.     The description of Fig. 9 is not very clear.

6.     Can the mRP metric proposed by the authors be used in other domains?

7.     Please check your manuscript again in respect of language. There are some spelling and grammatical mistakes.

Author Response

(The authors gave the same response as above.)
